# Fluctuations of psychological states on Twitter before and during COVID-19

**Johannes Massell, Roselind Lieb, Andrea Meyer, Eric Mayor** *

Division of Clinical Psychology and Epidemiology, Department of Psychology, University of Basel, Basel, Switzerland

* ericmarcel.mayor@unibas.ch

**Data Availability Statement:** The data underlying the results presented in the study are available on osf.io (https://osf.io/vpfky/).

## Abstract

The COVID-19 pandemic has been repeatedly associated with poor mental health. Previous studies have mostly focused on short time frames such as around the first lockdown periods, and the majority of research is based on self-report questionnaires. Less is known about the fluctuations of psychological states over longer time frames across the pandemic. Twitter timelines of 4,735 users from London and New York were investigated to shed light on potential fluctuations of several psychological states and constructs related to the pandemic. Moving averages are presented for the years 2020 and 2019. Further, mixed negative binomial regression models were fitted to estimate monthly word counts for the time before and during the pandemic. Several psychological states and constructs fluctuated heavily on Twitter during 2020 but not during 2019. Substantial increases in levels of sadness, anxiety, anger, and concerns about home and health were observed around the first lockdown periods in both cities. The levels of most constructs decreased after the initial spike, but negative emotions such as sadness, anxiety, and anger remained elevated throughout 2020 compared to the year prior to the pandemic. Tweets from both cities showed remarkably similar temporal patterns, and there are similarities to reactions found on Twitter following other previous traumatic events.

## Introduction

For over a year now, the world has been in the grip of COVID-19, which the World Health Organization (WHO) officially declared a pandemic on March 11, 2020. With over 175 million confirmed cases and 3.7 million confirmed deaths as of June 13, 2021 [1], it is now considered the biggest global crisis in recent history. To prevent the virus from spreading exponentially, most if not all countries introduced countermeasures such as the mandatory wearing of face masks, policies to enforce social distancing, and lockdowns during critical periods. Although the pandemic has been primarily a physical health crisis, the accompanying lockdowns were consistently associated with reduced mental well-being. These associations (e.g., symptoms of depression, anxiety, and stress) were observed in the general population in countries all over the world such as China [2, 3], Iran [4], Italy [5, 6], Germany [7], Switzerland [8], and the United Kingdom [9–11]. Systematic reviews and meta-analyses published in 2020 and early 2021 confirmed these early findings [12–16].

**Funding:** We thank the Publication Fund of the University of Basel for Open Access for funding the article processing charge of this article.

**Competing interests:** The authors have declared that no competing interests exist.

Although the existing literature demonstrates these associations for specific and limited time frames such as around the first lockdown periods, an understanding of the longitudinal patterns of psychological states across the pandemic is still lacking. Further, most previous studies relied on data stemming from self-report questionnaires, which are tied to a well-known set of biases (e.g., recall and non-response biases) [17]. The words a person uses on a daily basis, on the other hand, offer an entirely different avenue of research. Thus, a promising approach that compensates for some of these shortcomings is to look at data created organically without interference by an investigator. Over the last decade and a half, social media platforms such as Facebook and Twitter have steadily gained in popularity. Micro-blogs and social networking sites allow users to share updates about their lives, have conversations with each other, and edit content in real time [18]. Twitter has over 330 million users, with over 40% using the platform on a daily basis, producing half a billion tweets every day (https://www.omnicoreagency.com/twitter-statistics/). With this immense amount of data so readily available, it has become easier than ever to investigate the thoughts and behaviors of a substantial number of people in an unobtrusive and inexpensive manner.

In the past, social media has been successfully exploited to investigate psychological states during and after catastrophic events [19, 20]. Jones and Silver [21] demonstrated a sharp increase in anxiety on Twitter on the day of the false missile alert in Hawaii in 2018 and elevated anxiety levels for another week, even after the alert had turned out to be a false alarm. Lin et al. [22] used Twitter data to measure anxiety, anger, and sadness after the 2015 terrorist attacks in Paris. Looking at tweets following the Sandy Hook Elementary School shooting, Dore et al. [23] demonstrated that the use of sadness words decreased and the use of anxiety words increased with time and spatial distance. In a series of studies, Jones and colleagues investigated psychological states of Twitter users after mass violence on U.S. campuses and reported increased negative emotion expression after the attacks in Isla Vista [24] and San Bernardino [25] and acute stress during several campus lockdowns [26].

So far, only a handful of research projects have used social media to investigate the association between the COVID-19 pandemic and the psychological states of its users. Li et al. [27] investigated the association between the COVID-19 epidemic declaration in China and the psychological states of active Weibo (the Chinese equivalent of Twitter) users. They demonstrated an increase in anxiety, depression, indignation, and sensitivity to social risk as well as a decrease in positive emotions and life satisfaction during the week after the declaration. Su et al. [28] investigated the association between the COVID-19-induced lockdowns and the psychological states of Weibo users in Wuhan and Twitter users in Lombardy. They demonstrated an increased focus on "home" and a higher level of cognitive processing in both cities, decreased stress levels and increased attention to leisure in Lombardy, and increased attention to group, religion, and emotions in Wuhan. Zhang et al. [29] trained a machine learning model to detect signs of depression in users' Twitter content. They applied their model to Twitter feeds of randomly sampled users from three U.S. states and were able to show significant increases in depression signals between March and May 2020. Guntuku et al. [30] investigated several markers of mental health between March 13 (the day of the national declaration of emergency in the U.S.) and May 6, 2020, and demonstrated lower sentiment, higher stress, consistently higher anxiety, and an increase in loneliness in 2020 when compared to estimates from the same period in 2019. Li et al. [31] demonstrated a strong association between symptoms of psychological stress and the steadily increasing number of COVID-19 cases. Ashokkumar and Pennebaker [32] investigated fluctuations of American and Canadian Reddit users' psychological states during the first three months of the pandemic. They demonstrated increased anxiety levels and decreases in positive emotion and anger before the first lockdowns

were announced and increased levels of anxiety and sadness during the first lockdowns, imply-
ing substantial changes in psychological states during the early stages of the pandemic.

To summarize, the pandemic and accompanying lockdowns have consistently been shown
to be associated with increases in negative psychological states and decreases in positive psy-
chological states at specific points in time during 2020, such as around the first lockdown peri-
ods. How these psychological states further develop with time, how strongly they fluctuate
during the extended course of the pandemic, and whether they differ from periods before the
pandemic have not yet been investigated in depth. The present study set out to investigate
potential fluctuations of psychological states of Twitter users throughout the first year of the
pandemic (2020) and, for comparison, the previous year (2019). To achieve this goal, we col-
lected and analyzed the Twitter timelines of 4,735 users in London, England and New York.
We chose these two locations since both have English as their main language and both had
comparable first lockdown periods, which made them ideal candidates for our investigation.
We focused primarily on affective psychological states such as sadness, anxiety, and anger. Fur-
ther, we looked at constructs relating to more general aspects of life during the pandemic such
as home, work, and leisure.

## Methods

### Human subjects research

Review board approval was not necessary as the study relied on publicly available data that
users agreed in advance could be used for research in their license agreement with Twitter.

### Data source

To allow for future replication and comparison, we chose to source our data from the Mega-
COV data set [33], which consists of almost 1.5 billion tweets in over 100 languages from over
one million unique users in 268 countries. What sets this data set apart from other available
alternatives is the unfiltered nature of the collected tweets. We wanted to show the develop-
ment of psychological states of Twitter users within the whole Twitter sphere, not only in a
COVID-19-related or in any other way filtered subset of tweets, which made the Mega-COV
data set a suitable source.

### Rehydration procedure

We retrieved (*rehydrated* in Twitter jargon) the content of the tweets and associated informa-
tion such as user ID and location from the tweet IDs included in the Mega-COV data set for
January 2020 (our baseline for further comparisons in 2020). After rehydrating the slightly
over 73 million tweet IDs for January 2020 and filtering them by location (containing either of
the terms "London" or "New York") and by language (containing the term "en"), we were left
with 211,536 tweets from 3,262 unique users in London and 162,992 tweets from 2,007 unique
users in New York. We used Python 3.7.7 [34] for this initial rehydration and filtration pro-
cess, as it is well suited to handle large amounts of data in the JSON format.

In a second step, we rehydrated the complete Twitter timelines of these identified users. A
Twitter timeline can store up to 3,200 tweets, so depending on the frequency with which a user
produces content, the timelines can reach several years into the past. The second rehydration
procedure was conducted in R 4.0.2 [35] with the package rtweet [36]. After rehydration of the
user timelines, which took place during the first two weeks of January 2021, our data set con-
sisted of 6,918,209 tweets from 2,945 unique users in London and 4,392,801 tweets from 1,790
unique users in New York. The decrease in the number of users between the two rehydration

steps is attributed to revocation of public accessibility that Twitter users can enforce at any time (reasons for non-retrieval: account deleted or suspended, privacy settings).

## Preprocessing and filtering

Preprocessing was performed with preprocessor 0.6.0, a preprocessing library for tweets written in Python (https://github.com/s/preprocessor). We removed URLs, hashtags, mentions, reserved words (i.e., RT, FAV), emojis, smileys, and numbers from the tweets in our data set. Removing all tweets that were tagged as retweets reduced the sample size to 4,886,911 tweets for London and 3,129,183 tweets for New York. Further, removing all the tweets that had a word count of zero after preprocessing (e.g., tweets consisting of only a URL or an emoji) left us with 4,556,995 tweets for London and 2,905,928 tweets for New York. Finally, removing any tweet that was posted before January 1, 2019, or after December 31, 2020, left us with our final sample sizes of 1,314,414 tweets for 2020 and 762,170 tweets for 2019 for London, and 886,926 tweets for 2020 and 467,413 tweets for 2019 for New York.

## Linguistic Inquiry and Word Count (LIWC)

The words that make up tweets serve as indicators of the psychological states of interest. For our text analyses, we used Linguistic Inquiry and Word Count (LIWC), a simple but well-established and validated computerized text analysis method. LIWC features a set of predefined dictionaries, which allows the researcher to count words in psychologically meaningful categories [37]. LIWC has been used in the literature for the psycholinguistic analysis of text in a wide variety of research topics such as mood and anxiety disorders [38–40], psychological stress [41], and the relationship between emotions and well-being [42]. Originally designed for the analysis of more classic forms of text, LIWC has been successfully used for the psycholinguistic analysis of social media content. Specifically, it has proved to be useful in areas such as negative emotions following violence on U.S. campuses [24], work-related stress and emotions [43], and the differentiation between political orientations [44].

We used the LIWC2015 software for macOS [45] to calculate scores for the dictionary categories for each tweet. LIWC calculates scores for over 80 linguistic, psychological, and topical categories. As indicators of our psychological variables of interest, we used the following LIWC categories (with example words in parentheses): sadness (crying, grief, sad), anxiety (worried, fearful), anger (hate, kill, annoyed), negative emotion (hurt, ugly, nasty), positive emotion (love, nice, sweet), work (job, majors, xerox), leisure (cook, chat, movie), home (kitchen, landlord), and health (clinic, flu, pill).

Foremost we were interested in emotions during the pandemic as they entail a strong relationship with mental health. We thus included the LIWC categories sadness, anxiety, and anger (discrete emotions) as well as negative and positive emotion (generic emotions). We were further interested in the health category, as we anticipated the global health crisis that constitutes the pandemic might have had an impact on Twitter users' posts (e.g., reflecting fear for themselves or loved ones, or describing health issues). Covid and the stay-at-home mandates affected everyone's work routines, especially during the lockdowns, leading us to include the work and home categories. Leisure activities such as team sports were not possible during the early days of the pandemic. Again, we anticipated that these aspects of the pandemic would be reflected in the content of posts on Twitter. In sum, we chose the categories that, in our opinion, were most closely related to the pandemic and associated changes to society.

By default, the calculated LIWC category scores represent the proportion of in-category words compared to all words in any given piece of text. The different LIWC categories have different base rates. Reference values for our categories of interest within tweets can be found

in the LIWC2015 development manual [46]. They are based on an analysis, conducted with a computer program [47], of how often words belonging to the different LIWC categories can be found on average in various sources of text such as novels, blog posts, and tweets. The reference values of our relevant LIWC category scores for tweets are sadness (0.43), anxiety (0.24), anger (0.75), negative emotion (2.14), positive emotion (5.48), work (2.16), leisure (2.11), home (.43), and health (0.54).

## Moving averages

To present fluctuations of psychological states across time graphically, we calculated moving averages of the observed and unaltered LIWC scores of relevance. We aggregated the data by calculating mean LIWC scores for each user and day combination within a year and city. This step caps the influence that each Twitter user can have, regardless of the individual tweet count. For each day, a mean LIWC score was then calculated for the day itself and the 28 adjacent days. The 29-day time frame was chosen as it allowed us to showcase broader trends over the year without suffering from the noise that shorter time frames introduce. Selecting the 14 days prior to and after each day to calculate these moving averages provided the further benefit of capturing an equal proportion of weekdays to weekends.

## Transformation and modeling

LIWC scores calculated from tweets, especially those from categories belonging to the negative affective spectrum, tend have a vast excess of zeros. This was reflected by heavily right-skewed distributions of all dependent variables. Hence, to statistically compare fluctuations of psychological states across months, years and cities, we proceeded to analyze the data with mixed negative binomial (NB) regression models. These models allowed us to obtain measures of uncertainty (95% confidence intervals), despite the excess zeros and thus heavily right-skewed distributions of the dependent variables. NB and zero-inflated negative binomial (ZINB) models are recommended for distributions with excess zeros and high dispersion, whereas Poisson and zero-inflated Poisson (ZIP) models are recommended for distributions with excess zeros and low dispersion [48]. Indeed, preliminary benchmarks confirmed that NB and ZINB models outperformed Poisson and ZIP models for our data set of tweets, which did contain excess zeros and high dispersion. However, given that NB and ZINB models performed very similarly, as expressed by different model fit indices (Akaike information criterion, Bayesian information criterion, and log-likelihood), we chose the simpler NB models, thereby omitting zero-inflation terms.

Since mixed NB regression models deal with counts (ordinally scaled data with a minimum of 0) as outcomes, our dependent variables (LIWC category scores) had to be converted from proportions to counts. This was achieved by dividing LIWC scores by 100 and then multiplying them by the word count of the respective tweet. We set up a separate mixed NB regression model for each construct by year and city combination. Each model contained the factor month (12 levels for each month of the year) and the logarithm of the word count as covariate to control for the length of the tweets. In addition, each model contained a random intercept to account for the hierarchical structure of the data (tweets within persons). The models were set up in such a way that all months (February to December) were compared to a reference month, which was January of the corresponding year. We decided on January serving as the reference month because the focus of this study was on fluctuation across months within two full years. A total of 36 mixed NB regression models (nine dependent variables representing psychological states or relevant constructs by two years and two cities) were fitted. All models were fitted in R 4.0.3 with the package glmmTMB [49].

### Testing for fluctuation

To establish whether our dependent variables varied over the 12 months for a particular year in a city (omnibus test), we performed chi-square tests. To this end we compared model deviances between each full model including the fixed effect month and a control model omitting the fixed effect month, with the models being similar in every other regard. If a full model fit the data better than the corresponding control model, as assessed by the deviance change statistic and corresponding $p$ value, we reported the presence of fluctuation for a particular dependent variable by year and city combination. The significance threshold used for the chi-square tests was $p < 0.001$.

### Comparison of means

Statistically significant differences between estimated means were determined visually, by comparing the overlap of the corresponding 95% confidence intervals or the lack thereof. The use of confidence intervals, as opposed to reliance on simple $p$ values, is strongly recommended and described as "generally the best reporting strategy" in the seventh edition of the *Publication Manual of the American Psychological Association* [50]. The confidence intervals around the estimated monthly means depicted in the manuscript represent ranges of plausible values for monthly means and we can be 95% confident that these ranges include the corresponding population mean. We chose to be conservative with this approach by reporting differences in the manuscript only where the confidence intervals themselves do not overlap at all, which corresponds to $p$ values $< 0.01$ in the classic null hypothesis significance testing framework [51].

### Robustness check

To check the robustness of our conclusions, we additionally performed non-parametric bootstraps on monthly mean LIWC scores and compared the corresponding bootstrapped 95% confidence intervals. LIWC scores were pre-aggregated for each user by day combination within a year and city. This step caps the influence that each Twitter user can have on a given monthly mean regardless of individual tweet count and mimics the aggregation procedure used during the NB modeling process, improving comparability between the two methods. Non-parametric (because of the unmet assumption of normality) bootstrapped monthly means and 95% confidence intervals with 10,000 iterations each were calculated for every category by month and year combination.

### Data reporting

We are sharing our data set in two versions. The first version contains LIWC scores for each tweet posted in 2019 and 2020. With this data set, all analyses, figures, and tables can be replicated. The second version contains aggregated counts for each variable of interest. With this version of the data set the model fits can be replicated without having to perform the intermediate steps. Tweet content is omitted and user IDs are anonymized. Thus, all analyses are replicable while assuring privacy for the included Twitter users.

## Results

### Descriptive statistics

Twitter users in London published on average 446.8 (SD = 577.3) tweets per person in 2020 and 279.8 (SD = 311.3) tweets per person in 2019. A similar pattern emerged in New York with users publishing on average 496.0 (SD = 593.6) tweets per person in 2020 and 290.5

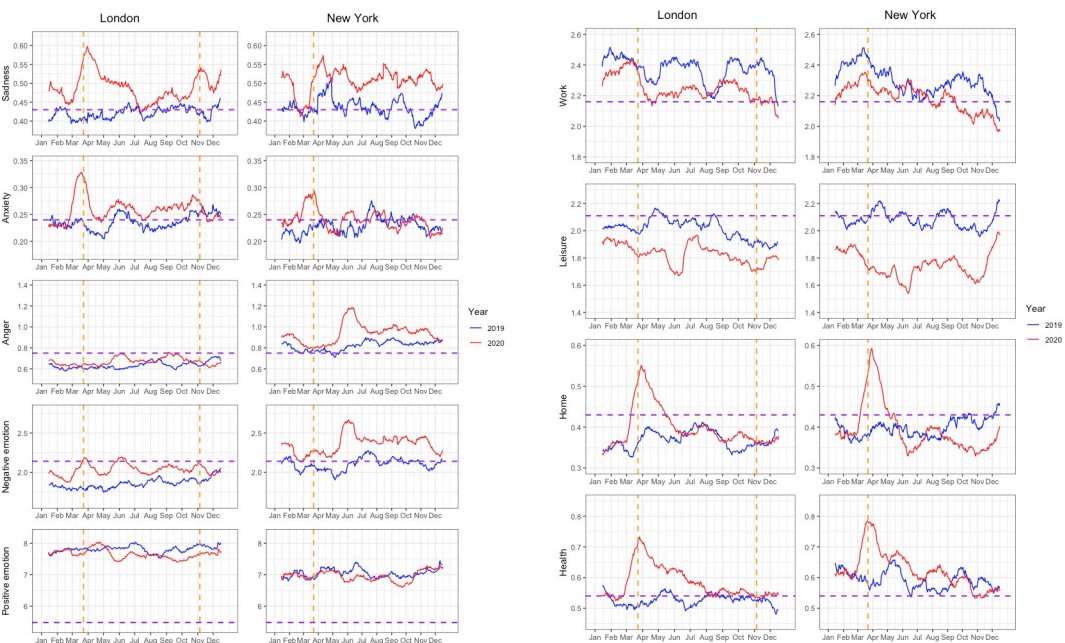

**Fig 1. a** Equally weighted 29-day moving averages of LIWC scores. *Note*. Orange dotted lines show the beginnings of the first (London and New York) and second (London only) lockdown; purple dotted lines show the Linguistic Inquiry and Word Count (LIWC) reference values for tweets; LIWC scores represent percentages of total in-category words. **b** Equally weighted 29-day moving averages of LIWC scores. Note. Orange dotted lines show the beginnings of the first (London and New York) and second (London only) lockdown; purple dotted lines show the Linguistic Inquiry and Word Count (LIWC) reference values for tweets; LIWC scores represent percentages of total in-category words.

(SD = 323.3) tweets per person in 2019. Monthly means plus standard deviations for each LIWC category score of interest are presented separately for London and New York for 2019 and 2020 (S1–S4 Tables).

## Moving averages of LIWC scores

Fig 1A and 1B depict the 29-day moving averages of each LIWC score for tweets published by users from our sample throughout the years 2020 and 2019, as well as the reference values reported by Pennebaker and colleagues [46]. These plots merely allow for a quick visual inspection of the LIWC scores across the entire study period and are not used to guide our interpretations, given the lack of measures of uncertainty.

## Estimated monthly means of in-category words

Twitter users in London published on average 618.0 (SD = 1213.76) words per month per person in 2020 and 396.3 (SD = 588.70) words per month per person in 2019. In New York, Twitter users published on average 652.8 (SD = 1190.83) words per month per person in 2020 and 409.5 (SD = 640.87) words per month per person in 2019. Fig 2A and 2B depict the estimated monthly means of in-category words for each LIWC category for an average user during the years 2020 and 2019, for each city. Given the substantial difference in total word counts between the years 2020 and 2019, the estimated monthly means of in-category words for each LIWC category is presented graphically for a fixed natural logarithm of total word count to ensure a "fair" comparison. Thus, for London and for New York, the total word counts were fixed to the 2020 means of ln(618) and ln(653), respectively.

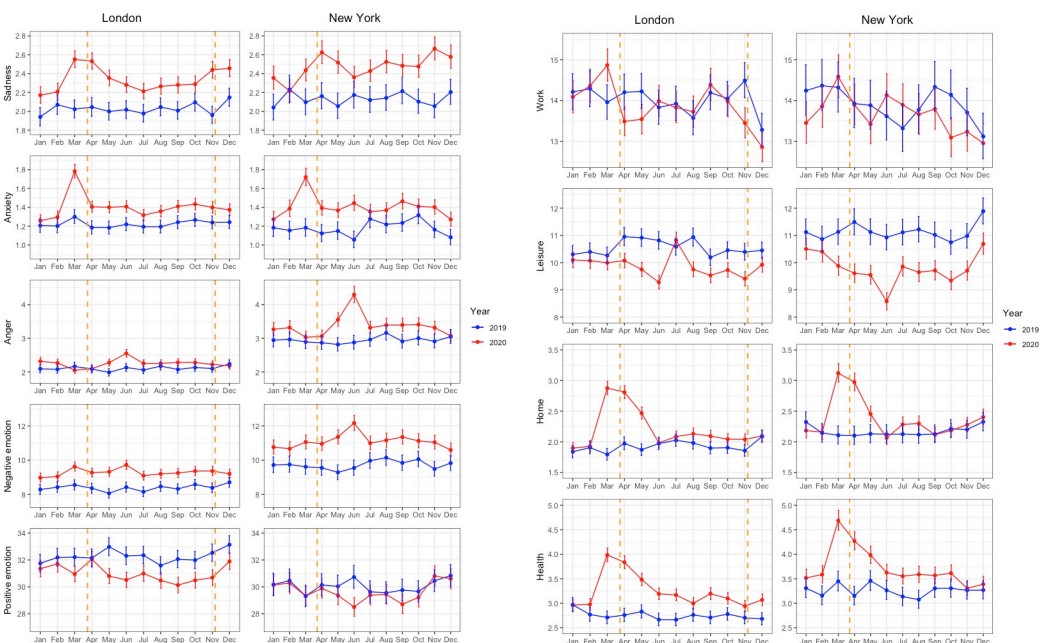

**Fig 2. a** Estimated monthly means of in-category word count for each LIWC category for an average user. *Note.* Orange dotted lines show the beginnings of the first (London and New York) and second (London only) lockdown. Estimates are based on mixed negative binomial regression models. The natural log of total word count was fixed to each city's 2020 mean. Error bars denote 95% confidence intervals. **b** Estimated monthly means of in-category word count for each LIWC category for an average user. *Note.* Orange dotted lines show the beginnings of the first (London and New York) and second (London only) lockdown. Estimates are based on mixed negative binomial regression models. The natural log of total word count was fixed to each city's 2020 mean. Error bars denote 95% confidence intervals.

## Testing for fluctuation

Chi-square tests we used to assess whether estimated monthly mean in-category word counts fluctuated across the 12 months within a year and city are presented in S5 and S6 Tables. The main comparisons, presented below, are based on monthly mean in-category word counts estimated by the mixed NB regression models. The corresponding estimated 95% confidence intervals were used to guide all interpretations apart from general fluctuation over the months within a year. Summaries of the individual mixed NB regression models are presented in S7–S15 Tables.

**Sadness, anxiety, anger, and negative emotion.** In London and New York, *sadness* fluctuated across months in 2020 but not in 2019 (see S5 and S6 Tables). For almost the entire year 2020, elevated levels of sadness were observed compared to 2019. In both cities, sadness increased early in 2020 before reaching a peak around the start of the first lockdown period. In London, sadness then decreased during the second quarter before increasing again during the third quarter and finally reaching a second peak around the start of the second lockdown period. In contrast to London, sadness in New York was consistently elevated throughout the year 2020 and did not show a clear dip during the summer months. Similarly, a second peak in sadness toward the end of 2020 was observed for New York as statewide restrictions were put in place. No such patterns were observed for 2019 for either city (see Fig 2A).

In London, *anxiety* fluctuated across months in 2020 but not in 2019, whereas in New York, anxiety fluctuated across both years, though less so in 2019 compared to 2020 (see S5 and S6 Tables). In both cities, elevated levels of anxiety persisted throughout the year 2020 compared to 2019 for most months. Also in both cities, anxiety increased in early 2020, reached a peak before the first lockdown period, and then returned back to baseline with no

additional prominent spikes throughout the rest of 2020. No such patterns were observed for 2019 for either city (see Fig 2A).

*Anger* fluctuated in both cities across the months in 2020 but not in 2019 (see S5 and S6 Tables). Anger started to increase around the beginning of the first lockdown period, especially in New York, and reached a peak in June before declining again to pre lockdown levels from July onward. Values for anger were only slightly elevated in 2020 compared to 2019 outside of the peak period in June. Twitter users in New York generally posted far more angry words than users in London during both years (see Fig 2A).

For *negative emotion*, values fluctuated across the months in 2020 and 2019 in both cities (see S5 and S6 Tables). Increased levels of negative emotion were observed throughout the entire year of 2020 compared to 2019. A peak in negative emotion was apparent around June of 2020, especially in New York, but not in 2019 for the same period. Twitter users in New York generally posted more words from the negative emotion category than users in London during both years (see Fig 2A).

**Positive emotion.**  *Positive emotion* fluctuated over the months in both cities and both years (see S5 and S6 Tables). Decreased levels of positive emotion in 2020 compared to 2019 were observed in London following the onset of the lockdown from May 2020 up to the end of the year. In New York, in contrast, values for positive emotion were comparable between the years except for June, where values were higher in 2019 than in 2020. Twitter users in New York generally posted fewer words from the positive emotion category than users in London during both years (see Fig 2A).

**Work, leisure, home, and health.**  The use of *work-related words* fluctuated over the months in both cities and both years (see S5 and S6 Tables). Observed levels were comparable between 2020 and 2019 in both cities except for the peak in London in March 2020 shortly before the first lockdown, which was not observed for 2019 for the same month (see Fig 2B).

As for work-related words, the use of *leisure-related words* fluctuated over the months in both cities and both years (see S5 and S6 Tables). The use of leisure-related words decreased around the start of the first lockdown in both cities and stayed low for the remainder of 2020, compared to the year before the pandemic. The only exception concerned the month of July in London with comparable values between 2020 and 2019 due to a sudden increase in 2020 (see Fig 2B).

The use of *home-related words* fluctuated across months during both years in London, but only in 2020 in New York (see S5 and S6 Tables). Values for home-related words strongly increased in 2020 before the first lockdown in both cities, reaching a peak around the start of the first lockdown before slowly declining back to baseline. No such pattern was observed for 2019, and from June onward levels were almost identical for the two years in both cities (see Fig 2B).

Finally, values for *health-related words* followed a similar pattern to that of home-related words. Fluctuation among months was observed for 2020, but not for 2019, in both cities (see S5 and S6 Tables). Values for health-related words increased in 2020 before the first lockdown, reaching a peak around the start of the first lockdown before slowly declining back to baseline. No such pattern was observed in 2019. From June onward, values remained more or less constant in both cities up to the end of the year but became somewhat elevated in 2020 compared to 2019 (see Fig 2B).

## Robustness check

With few exceptions the results of our robustness check allow us to draw the same conclusions. We attribute the minor differences to the heterogeneity between proportions and counts,

aspects of the analytic methods themselves, and the degree to which extreme cases are controlled for. The aggregated results of the robustness check are presented in S1A and S1B Fig.

## Discussion

In our study, we set out to investigate fluctuations of psychological states and of constructs relating to general aspects of life based on Twitter users' tweets made in London and New York during 2020, the first year of the COVID-19 pandemic, and, for comparison, tweets made in the same locations during the previous year. Unusual fluctuations were observed during the year of the pandemic: Sadness increased in tandem with both lockdown periods, anxiety increased before the first but not before the second lockdown period, and anger increased with a delay toward the end of the first lockdown period. Negative emotions, including sadness, anxiety, and anger, were generally more prevalent in tweets during the year of the pandemic. Positive emotion and the use of leisure-related words, on the other hand, were less prevalent. The use of home- and health-related words increased in tandem with the first lockdown period, but this surge did not repeat for the second lockdown period toward the end of 2020. The Twitter landscapes of both cities showed similar patterns for the majority of the investigated constructs.

The largest fluctuations of psychological states in our sample occurred around the first lockdown periods in 2020, with anxiety peaking before lockdown, sadness, home-related words, and health-related words peaking during lockdown, and anger peaking after lockdown, before declining again. Patterns such as these are relatively common when looking at the relationship between the occurrence of traumatic events and subsequent expressions found in the news and on social media. Talking about traumatic experiences, or "social sharing," is a form of processing and coping that follows emotional upheaval [52]. Pennebaker and Harber [53] proposed a social stage model of collective coping that distinguishes between three stages. Immediately after a traumatic event during an emergency stage, people have the tendency to openly talk and think about their emotional experiences. This is followed by an inhibition stage in which people stop talking but keep thinking about the event in question. Last, an adaption phase is entered into in which people neither talk nor think about the event, which will have been processed by that time.

Comparing tweets published between March and May of 2020 and during a comparable period in 2019, Saha and colleagues reported similar patterns of spikes followed by declines and eventual plateauing of several expressions on Twitter symptomatic of psychosocial and mental health issues during the early stages of the pandemic [54]. They attributed the development of this pattern to either habituation or successful supportive policy measures. Ashokkumar and Pennebaker [32] reported similar patterns: a surge in anxiety and a drop in positive emotion before the first lockdown, a spike in anxiety and increased sadness at the beginning of the lockdown, and stabilized but not abated psychological states six weeks after the lockdown had ended. Although most of the psychological states investigated in our study exhibited the greatest fluctuation around the first lockdown, a pattern that is reported in a plethora of previous research into specific limited stages of the COVID-19 pandemic, this does not paint the full picture for all of the constructs under investigation. Thus, in addition, sadness, anxiety, anger, and negative emotions showed persistently elevated levels throughout 2020 compared to 2019, and positive emotion showed persistently reduced levels in tweets from London during most of 2020 compared to 2019. Sadness started to increase again around the beginning of the second lockdown, but anxiety did not. Twitter users seemed to have developed a form of habituation regarding anxiety, anger, home, and health. However, sadness never reached levels comparable to 2019 and showed signs of a second spike toward the end of the year.

The patterns that we observed are not exclusive to social media, however. In a sample of the general public and health care professionals in Wuhan, China, symptoms of anxiety and depression decreased after the lockdown was eased [55], although in that study, a time frame with no new infections was compared to the first hard lockdown at the very early stages of the pandemic. Further support for this pattern of spikes followed by steady declines comes from England. Fancourt et al. [56] reported the highest levels of anxiety and depression during the first week of the U.K. lockdown. Levels of anxiety and depression then steadily decreased over the following 20 weeks. The psychological states of Twitter users found in our research therefore seem reflective of respondents' answers in questionnaire-based studies.

Our study is not without limitations. First, LIWC was originally designed with more conventional pieces of text in mind, and tweets deviate considerably in this regard. Tweets often contain (internet) slang, emojis, hyperlinks, and attached media. Since for the purpose of our study such elements were removed, parts of the content of the investigated tweets must be considered missing. Emojis in particular transfer emotional content that can be complementary to that coded from text alone [57]. However, the inclusion of emojis in our data would have rendered our analyses much more complicated, with the integration of information provided by LIWC scores on the one hand, and the significance of emojis on the other. Such analyses were not within the scope of the present study but will be considered for potential follow-up projects.

Second, LIWC2015 does not take into account the context of words. It is therefore capable of detecting neither negation nor sarcasm. The Valence Aware Dictionary for Sentiment Reasoning (VADER) is a dictionary-based text analysis method able to take certain contextual elements of language such as negation detection into account in the computation of positive, negative, compound, and neutral sentiment scores [58]. We therefore conducted an additional VADER analysis on the same tweets that were used in our LIWC analysis, to check if the inability to detect contextual features such as negation was problematic for the interpretation of our LIWC analysis. The results of the VADER analysis were similar to the results of the negative and positive emotion categories reported for our LIWC analyses, suggesting that unaccounted-for contextual features did not make a noticeable difference. The results of this VADER analysis are reported in S2 Fig in the form of 29-day moving averages.

Sarcasm entails saying the opposite of what is meant and cannot be detected using LIWC. Automated sarcasm detection is currently being worked on with the help of sophisticated machine learning approaches [59], but current algorithms are still not capable of indicating what exact word is meant as the opposite of what is said, which is problematic as one tweet can contain both sarcastic and non-sarcastic words. We point out that sarcasm is relatively infrequent in tweets with only between 1% and 4% of tweets reported to be sarcastic in recent studies [60–62]. It is therefore very unlikely that sarcastic words in the tweets of our sample had a relevant impact on our results.

Third, while people from all walks of life use Twitter, the overall population of Twitter users cannot be considered representative of the general population and must instead be regarded as a "highly non-uniform sample of the population". An overrepresentation of densely populated areas in the United States with substantially more male than female users and an uneven distribution of ethnicity was reported in a 2011 study [63]. McCormick et al. [64], by contrast, although agreeing on a non-representative population and an uneven distribution of ethnicities, reported an even distribution between male and female users and a predominantly younger user population below the age of 50 from more urban than rural areas.

A further limitation is the difficulty in distinguishing between personal and business accounts on Twitter. The Chinese equivalent Weibo features an internal flag that declares an account as either a personal or a business account, but Twitter does not offer that feature. This

means that in our analyses both personal and business accounts were included. Since we are interested in psychological states, this problem dilutes our data set.

In the present study, we compared the entire first year of the pandemic (2020) with the previous year (2019). Although our investigation covered a substantial time span that, to our knowledge, at present remains unequalled in previous research, the period under investigation here still covered only the first wave and the beginning of the second wave of the pandemic. Looking at the course COVID-19 has taken so far; the first wave was almost insignificant in terms of infection rate and death toll compared to what followed from November 2020 to the present day. Although vaccination is underway around the globe, albeit at a mixed pace, and an end to the pandemic appears likely at some future point, there have been multiple additional waves and the pandemic continues to rage all over the world. It would therefore certainly be interesting for future studies to evaluate potential fluctuations of psychological states during 2021 and beyond.

In conclusion, psychological states on Twitter fluctuated heavily during the first year of the pandemic. Almost all constructs under investigation showed a markedly different temporal fluctuation in 2020 compared to 2019. Although the period around the first lockdown showed the highest fluctuation, the story does not end there. A certain form of habituation was observed for many of the constructs, but the taxing effects of the pandemic lingered on throughout the year, and sadness started to rise again in tandem with the second wave. Although causality cannot be demonstrated by our observational design, we nevertheless believe that our results provide further evidence of the negative effects of COVID-19 and accompanying lockdowns on the population's mental health. Our data suggest that the mental health crisis associated with this pandemic is ongoing, and additional attention and resources are therefore needed to mitigate further damage and to find the means to remedy the situation in the long term in responsible and sustainable ways.

## Supporting information

**S1 Table. Mean monthly LIWC scores for tweets from London during 2020.** *Note.* NegEmo = Negative emotion; PosEmo = positive emotion; Linguistic Inquiry and Word Count (*LIWC) scores represent percentages of total in-category words within a given text.* (DOCX)

**S2 Table. Mean monthly LIWC scores for tweets from London during 2019.** *Note.* NegEmo = Negative emotion; PosEmo = positive emotion; Linguistic Inquiry and Word Count (LIWC) scores represent percentages of total in-category words within a given text. (DOCX)

**S3 Table. Mean monthly LIWC scores for tweets from New York during 2020.** *Note.* NegEmo = Negative emotion; PosEmo = positive emotion; Linguistic Inquiry and Word Count (LIWC) scores represent percentages of total in-category words within a given text. (DOCX)

**S4 Table. Mean monthly LIWC scores for tweets from New York during 2019.** *Note.* NegEmo = Negative emotion; PosEmo = positive emotion; Linguistic Inquiry and Word Count (LIWC) scores represent percentages of total in-category words within a given text. (DOCX)

**S5 Table. Testing for fluctuations over the different months within a year, based on chi-square tests between the models with and without the factor month for London.** *Note.*

AIC = Akaike information criterion; BIC = Bayesian information criterion.
(DOCX)

**S6 Table. Testing for fluctuations over the different months within a year, based on chi-square tests between the models with and without the factor month for New York.** *Note.* AIC = Akaike information criterion; BIC = Bayesian information criterion.
(DOCX)

**S7 Table. Mixed negative binomial regression models predicting the monthly number of words belonging to the LIWC dictionary "Sadness".** *Note.* CI = confidence interval; ICC = intraclass correlation coefficient; LIWC = Linguistic Inquiry and Word Count; uid = user id; wc = word count.
(DOCX)

**S8 Table. Mixed negative binomial regression models predicting the monthly number of words belonging to the LIWC dictionary "Anxiety".** *Note.* CI = confidence interval; ICC = intraclass correlation coefficient; LIWC = Linguistic Inquiry and Word Count; uid = user id; wc = word count.
(DOCX)

**S9 Table. Mixed negative binomial regression models predicting the monthly number of words belonging to the LIWC dictionary "Anger".** *Note.* CI = confidence interval; ICC = intraclass correlation coefficient; LIWC = Linguistic Inquiry and Word Count; uid = user id; wc = word count.
(DOCX)

**S10 Table. Mixed negative binomial regression models predicting the monthly number of words belonging to the LIWC dictionary "NegEmo".** *Note.* CI = confidence interval; ICC = intraclass correlation coefficient; LIWC = Linguistic Inquiry and Word Count; uid = user id; wc = word count.
(DOCX)

**S11 Table. Mixed negative binomial regression models predicting the monthly number of words belonging to the LIWC dictionary "PosEmo".** *Note.* CI = confidence interval; ICC = intraclass correlation coefficient; LIWC = Linguistic Inquiry and Word Count; uid = user id; wc = word count.
(DOCX)

**S12 Table. Mixed negative binomial regression models predicting the monthly number of words belonging to the LIWC dictionary "Work".** Note. CI = confidence interval; ICC = intraclass correlation coefficient; LIWC = Linguistic Inquiry and Word Count; uid = user id; wc = word count.
(DOCX)

**S13 Table. Mixed negative binomial regression models predicting the monthly number of words belonging to the LIWC dictionary "Leisure".** *Note.* CI = confidence interval; ICC = intraclass correlation coefficient; LIWC = Linguistic Inquiry and Word Count; uid = user id; wc = word count.
(DOCX)

**S14 Table. Mixed negative binomial regression models predicting the monthly number of words belonging to the LIWC dictionary "Home".** *Note.* CI = confidence interval; ICC = intraclass correlation coefficient; LIWC = Linguistic Inquiry and Word Count;

uid = user id; wc = word count.
(DOCX)

**S15 Table. Mixed negative binomial regression models predicting the monthly number of words belonging to the LIWC dictionary "Health".** *Note*. CI = confidence interval; ICC = intraclass correlation coefficient; LIWC = Linguistic Inquiry and Word Count; uid = user id; wc = word count.
(DOCX)

**S1 Fig. Monthly LIWC scores with bootstrapped 95% confidence intervals.** *Note*. Orange dotted lines show the beginnings of the first (London and New York) and second (London only) lockdown; purple dotted lines show the Linguistic Inquiry and Word Count (LIWC) reference values for tweets; LIWC scores represent percentages of total in-category words; monthly means are from the original sample; error bars denote bootstrapped 95% confidence intervals aggregated from 10,000 iterations.
(DOCX)

**S2 Fig. Equally weighted 29-day moving averages of VADER scores.** *Note*. Orange dotted lines show the beginnings of the first (London and New York) and second (London only) lockdown; neg = negative; pos = positive; neu = neutral; VADER = Valence Aware Dictionary for Sentiment Reasoning.
(DOCX)

## Author Contributions

**Conceptualization:** Johannes Massell, Roselind Lieb, Andrea Meyer, Eric Mayor.

**Data curation:** Johannes Massell.

**Formal analysis:** Johannes Massell, Andrea Meyer, Eric Mayor.

**Investigation:** Johannes Massell.

**Methodology:** Johannes Massell, Andrea Meyer, Eric Mayor.

**Project administration:** Johannes Massell, Roselind Lieb, Eric Mayor.

**Resources:** Roselind Lieb.

**Software:** Johannes Massell.

**Supervision:** Roselind Lieb, Andrea Meyer, Eric Mayor.

**Validation:** Johannes Massell, Roselind Lieb, Andrea Meyer, Eric Mayor.

**Visualization:** Johannes Massell.

**Writing – original draft:** Johannes Massell.

**Writing – review & editing:** Roselind Lieb, Andrea Meyer, Eric Mayor.

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
