## [Decision Letter · Decision Letter 0]

15 Jun 2022

PONE-D-21-22931Fluctuations of Psychological States on Twitter Before and During COVID-19PLOS ONE

Dear Dr. Mayor,

Thank you for submitting your manuscript to PLOS ONE. After careful consideration, we feel that it has merit but does not fully meet PLOS ONE’s publication criteria as it currently stands. Therefore, we invite you to submit a revised version of the manuscript that addresses the points raised during the review process.

Specifically, Reviewer 2 worries about the bias inherent in all studies using social media. The authors should explain how they estimate this bias, and how they eliminate or minimize it. Reviewer 1 also raised statistical concerns on the method used. Please provide justifications for the choice of method, and also check that the results are robust.

We look forward to receiving your revised manuscript.

Kind regards,

Siew Ann Cheong, Ph.D.

Academic Editor

PLOS ONE

Journal Requirements:

2. We noted in your submission details that a portion of your manuscript may have been presented or published elsewhere. "Abdul-Mageed et al., 2021". Please clarify whether this publication was peer-reviewed and formally published. If this work was previously peer-reviewed and published, in the cover letter please provide the reason that this work does not constitute dual publication and should be included in the current manuscript.

Reviewers' comments:

Reviewer's Responses to Questions

**Comments to the Author**

1. Is the manuscript technically sound, and do the data support the conclusions?

Reviewer #1: Yes

Reviewer #2: Partly

2. Has the statistical analysis been performed appropriately and rigorously? 

Reviewer #1: No

Reviewer #2: Yes

3. Have the authors made all data underlying the findings in their manuscript fully available?

Reviewer #1: Yes

Reviewer #2: No

4. Is the manuscript presented in an intelligible fashion and written in standard English?

Reviewer #1: Yes

Reviewer #2: Yes

5. Review Comments to the Author

Reviewer #1: The authors present a comprehensive study of the psycho-social impact of COVID using Twitter. The manuscript is well-written, and the design is a useful conceptual replication of past work. While I believe the modeling and conclusions are likely sound, the explanation of the analysis needs more detail before I could make final recommendation. With suitable revisions to make the analysis clearer, the manuscript would likely be a great candidate for publication.

While I appreciate that this is a complex statistical problem, I am not convinced that the chosen analytic method is the best choice as it is very difficult to interpret. I am not suggesting that the authors use a different method entirely, but I would like to see (1) more justification for why this method was chosen, and (2) a robustness check of some sort. For example, the authors could calculate bootstrapped means (and confidence intervals) for each month, then do simple pair-wise comparisons to determine if the same conclusions would be drawn with a different analytic method.

The authors make claims that there is more variance in 2020 compared to 2019, but the analyses don’t clearly support that conclusion. An analysis of the standard deviation (or some other metric of variance) would be needed to fully support that claim.

The interpretation of the regression models could be clearer. First, why was January chosen as the only reference month? This choice would limit the interpretation to just whether each month differed from January. February would seem to be a better reference month since the lockdowns were in March.

Additionally, it is unclear exactly which statistics from the models the authors are using to support their conclusions. The graphs in Figures 1a and 1b don’t seem to support some of the conclusions. The authors need to be explicit about how they are interpreting the models and drawing conclusions. Furthermore, the authors need to explain how the models relate to the graphs. Relatedly, how was ‘significance’ determined?

Minor Comments:

The inclusion of the emotion LIWC categories is obvious, but the authors should include a paragraph justifying and explaining why they chose to examine the LIWC categories they did.

This is only suggestion, but the authors might consider also including an analysis using VADER (in addition to LIWC). The authors correctly point out the LIWC is not ideal for analyzing Twitter. VADER would allow for the study of positive and negative emotion but would not have the same limitation as VADER was validated with social media data.

Since there are so many graphs, it would be useful for all of them to have axis labels to make them clearer.

Reviewer #2: This is very interesting work. I would like to point out that in similar textual analysis that I have seen on social media, negation detection and sarcasm handling played a huge bias. I would appreciate the authors views or better yet, analysis taking these factors into account

6. PLOS authors have the option to publish the peer review history of their article (what does this mean?). If published, this will include your full peer review and any attached files.

Reviewer #1: **Yes: **Kayla Jordan

Reviewer #2: No

---

## [Author Response · Author response to Decision Letter 0]

13 Aug 2022

The response to reviewers is provided in the attached word document Response to Reviewers.

---

## [Decision Letter · Decision Letter 1]

9 Nov 2022

Fluctuations of Psychological States on Twitter Before and During COVID-19

PONE-D-21-22931R1

Dear Dr. Mayor,

We’re pleased to inform you that your manuscript has been judged scientifically suitable for publication and will be formally accepted for publication once it meets all outstanding technical requirements.

Kind regards,

Siew Ann Cheong, Ph.D.

Academic Editor

PLOS ONE

Additional Editor Comments (optional):

Reviewers' comments:

Reviewer's Responses to Questions

**Comments to the Author**

1. If the authors have adequately addressed your comments raised in a previous round of review and you feel that this manuscript is now acceptable for publication, you may indicate that here to bypass the “Comments to the Author” section, enter your conflict of interest statement in the “Confidential to Editor” section, and submit your "Accept" recommendation.

Reviewer #1: All comments have been addressed

Reviewer #3: (No Response)

2. Is the manuscript technically sound, and do the data support the conclusions?

Reviewer #1: (No Response)

Reviewer #3: Partly

3. Has the statistical analysis been performed appropriately and rigorously? 

Reviewer #1: (No Response)

Reviewer #3: I Don't Know

4. Have the authors made all data underlying the findings in their manuscript fully available?

Reviewer #1: (No Response)

Reviewer #3: Yes

5. Is the manuscript presented in an intelligible fashion and written in standard English?

Reviewer #1: (No Response)

Reviewer #3: Yes

6. Review Comments to the Author

Reviewer #1: (No Response)

Reviewer #3: Thanks for inviting me to review this paper, it is an interesting one on the Fluctuations of Psychological States on Twitter Before and During COVID-19. There are some comments and suggestions for further improvement:

1. Research question(s) could be listed clearly to present the aim and objective of this study.

2. The authors use LIWC for collecting different mental states from Twitter, however the keywords choice(eg: crying, grief, sad) and the categories(eg: sadness) choice need literature to support them. eg: "LIWC categories (with example words in parentheses): sadness (crying, grief, sad), anxiety (worried, fearful), anger (hate, kill, annoyed), negative emotion (hurt, ugly, nasty), positive emotion (love, nice, sweet), work (job, majors, xerox), leisure (cook, chat, movie), home (kitchen, landlord), and health (clinic, flu, pill)".

3. The author stated that the "results provide negative effects of COVID-19 and accompanying lockdowns on the population’s mental health"(page 25). however, this paper only demonstrates correlation, but not causal relation (impact influence).

4. The authors suggest that "additional attention and resources are therefore needed in order to mitigate further damage and to find the means to remedy the situation in the long term in responsible and sustainable ways." (page 25). Further detailed constructive comments or practical suggestions could be added according to the findings in the paper to strengthen the conclusion and discussion.

7. PLOS authors have the option to publish the peer review history of their article (what does this mean?). If published, this will include your full peer review and any attached files.

Reviewer #1: **Yes: **Kayla Jordan

Reviewer #3: No

---

## [Editor Report · Acceptance letter]

16 Nov 2022

PONE-D-21-22931R1 

Fluctuations of Psychological States on Twitter Before and During COVID-19 

Dear Dr. Mayor:

I'm pleased to inform you that your manuscript has been deemed suitable for publication in PLOS ONE. Congratulations! Your manuscript is now with our production department. 

Kind regards, 

on behalf of

Dr. Siew Ann Cheong 

Academic Editor

PLOS ONE